# Supplementation with Potato Protein Concentrate and *Saccharomyces boulardii* to an Antibiotic-Free Diet Improves Intestinal Health in Weaned Piglets

**DOI:** 10.3390/ani15070985

**Published:** 2025-03-29

**Authors:** Tércia Cesária Reis de Souza, Gerardo Mariscal Landín, Ulisses Moreno Celis, Teresita Hijuitl Valeriano, José Guadalupe Gómez-Soto, Christian Narváez Briones

**Affiliations:** 1Faculty of Natural Sciences, Autonomous University of Querétaro, Querétaro 76230, Mexico; ulisses.moreno@uaq.mx (U.M.C.); teresita.hijuitl@uaq.mx (T.H.V.); jose.gomez@uaq.mx (J.G.G.-S.); chnarvaezb@gmail.com (C.N.B.); 2National Institute of Agricultural and Livestock Forestry Research, National Center of Research in Animal Physiology, Ajuchitlán Colón, Querétaro 76280, Mexico; mariscal.gerardo@inifap.gob.mx

**Keywords:** piglets, occludins, intestinal permeability, functional foods, probiotics

## Abstract

The development of microbial resistance to antibiotics added to piglet feed has led to the search for alternatives to their use to ensure intestinal health during the first two weeks post-weaning, which is considered the most critical stage of this phase. The purpose of this work was to assess whether the consumption of two functional foods (potato protein concentrate and *Saccharomyces boulardii*), isolated or combined, could improve various aspects related to gastrointestinal health. The results obtained may provide nutritionists with new alternatives to improve piglet intestinal health post-weaning phase.

## 1. Introduction

After birth, piglets have a low capacity to digest and absorb nutrients of plant origin, and their digestive systems must therefore mature quickly to ensure survival [1]. The post-weaning period is thus critical for pig production, health, and performance. This has therefore generated substantial worldwide interest in mitigating post-weaning challenges and identifying nutritional alternatives, management, and disease prevention to positively influence the structure and function of the gastrointestinal tract (GIT), thereby improving production and health outcomes [2]. Stress at weaning, in addition to the presence of new pathogens, leads to neuroendocrine, immunological, and digestive complications [3].

Consequently, in the first 24–48 h after weaning, we observe changes in the intestine morphology of piglets; the destruction of intestinal barrier function; the atrophy of the intestinal mucosa, provoking a decrease in digestive and absorptive capacity; reduced feed intake; an increased diarrhea rate; and growth retardation [4]. Diarrhea is widely related to sudden changes in diet from milk to solid feed and gastrointestinal infections. Both factors promote rapid bacterial dysbiosis with an abrupt reduction in *Lactobacillus* spp., which contributes to the loss of intestinal structure [5]. Antibiotics have traditionally been used as growth promoters to combat gastrointestinal complications because their use in animal feed improves the animal growth rate and reduces the incidence of diseases and mortality [6].

To reduce antibiotic usage in animal production, a global effort has been undertaken [7], and feed-based antibiotics have been prohibited in several parts of the world. Although Mexico has a mandatory National Action Strategy against antimicrobial resistance, published in the Official Gazette of the Federation in June 2018, there is still no regulatory legislation that effectively controls antimicrobial usage. Meanwhile, international pressure has promoted research into alternatives to antibiotics in pig feed [7]. The use of potato protein concentrate (PP) in antibiotic-free diets had a positive effect on the ileal digestibility of nutrients in piglets, as well as improving the use of dry matter and energy throughout the GIT [8,9]. In addition to being a protein of high nutritional quality, it offers functional benefits for human health like antiallergic [10], antioxidant [11,12], and anticancer activity [11,13], which can also cover animal health. We are particularly interested in actions that can promote intestinal health in piglets on antibiotic-free diets, due to peptides’ antifungal and antimicrobial properties [14].

Yeasts have been used as probiotics in animal diets to reduce the use of growth-promoting antibiotics owing to their positive effects on productive criteria, intestinal microbiota, diarrhea incidence, the intestinal anti-inflammatory response, the health and integrity of the intestinal mucosa, and secretion of metabolites that improve the immune response [15], as well as their antiviral action [16]. The advantages of a single, early-life administration of *Saccharomyces boulardii* for the growth performance and fecal microbiota of piglets from birth until weaning have been reported [17]. *Saccharomyces boulardii* is used in animals feed because of its benefits in reducing diarrhea in young animals and improving body immunity, intestinal barrier function, and performance; it effectively reduces the colonization and translocation of pathogenic bacteria in the digestive tract of pigs, increases IgA antibody secretion, inhibits inflammation, and decreases mortality [18]. Another study with *Saccharomyces cerevisiae* var. *boulardii* RC009 observed improvements in zootechnical parameters in the post-weaning stage and an increase in the health status of the animals, indicating that it may be a promising alternative to prophylactic antibiotics [19]. PP and Sb therefore meet the definition of functional foods [20] because they are characterized by improving the intestinal health and therapeutic conditions of individuals when consumed in adequate quantities.

Although the term “gut health” in animals is not well defined, several items related to gut structure and function and microbial population have been used to describe gut health outcomes [21]. A healthy gut may include a healthy proliferation of intestinal epithelial cells, an integrated gut barrier function, a balanced gut microbiota, and a well-developed intestinal mucosa immunity [22]. An important function of a healthy small intestine is to serve as a physical barrier, facilitating the uptake of essential nutrients while limiting the migration of pathogenic molecules [21]. Intestinal barrier function is achieved primarily by regulating the synthesis of tight junction (TJ) proteins, such as occludins, which are critical for maintaining optimal gut health [23].

The objective of this study was to evaluate the effects of PP and *Saccharomyces boulardii* (Sb), either alone or in combination, on piglet performance, digestive organ weight, pH of digestive contents, the intestinal morphology of the villi and crypts, the presence of occludin proteins in jejunum and colon samples, coliform and *Lactobacillus* populations in feces, and the incidence and severity of diarrhea during the first two weeks post-weaning.

## 2. Materials and Methods

This study was approved by the Bioethical Committee of the Faculty of Natural Sciences of the Autonomous University of Querétaro (approval number: 63FCN2021) and was conducted at the experimental farm of CENID Physiology at the National Institute of Agricultural and Livestock Forestry Research, National Center of Research in Animal Physiology, Mexico. The pigs were treated according to the International Guiding Principles for Biomedical Research Involving Animals [24] and Official Mexican Standards and Regulations for the Technical Specifications of Production, Care, and Use of Laboratory Animals (NOM-062-ZOO-1999) [25].

### 2.1. Animals, Housing, Diets, and Management

A total of 132 hybrid piglets (Fertilis × Genetiporc) were weaned at 19.8 ± 1.6 days and weighed 6.2 ± 0.85 kg. They were then divided into five groups: two groups of 27 piglets and three groups of 26 piglets, each consuming one of the following experimental diets (Table 1): C− (negative control, without antibiotics, PP, or Sb), C+ (positive control, with antibiotics), SB (with Sb), PPC (with PP), or PPC-SB (with PP and Sb). The diets were formulated based on the Ideal Protein concept, using the standardized ileal digestibility of amino acids. All the diets met the nutritional requirements of the National Research Council (NRC) [26]. The PP and Sb were commercially available products donated by Grupo NUTEC^®^ (Querétaro, Mexico) and Lallemand Animal Nutrition (Montreal, Quebec, Canada), respectively. The antibiotic used was Linco-Spectin premix, containing 2.2 g of lincomycin and 2.2 g of spectinomycin (Zoetis, Mexico City, Mexico).

To homogenize the size of the piglets in the pens, minimize competition among them, and improve their well-being, the piglets were assigned to each diet according to a randomized complete block design, with their initial weight used as the blocking factor (large, medium, and small weight).

The piglets were moved to a weaning room with a controlled environment (30 °C during the first week post-weaning and 28 ± 2 °C during the second week). They were housed in pens raised 38 cm above the ground, measuring 115 cm in width and 150 cm in length, providing a total surface area of 1.7 m^2^. Each pen contained a nipple drinker, a feeder, and a plastic grid floor. Feed was offered three times a day at 8:00, 12:00, and 16:00, and feed intake was recorded daily. The piglets were weighed at weaning and at the end of weeks one and two post-weaning. Simultaneously, average daily feed intake (ADFI), average daily gain (ADG), and feed efficiency (FE) were determined. All the animals had ad libitum access to water.

### 2.2. Digestive Morphophysiological Characteristics

To evaluate the effect of functional food intake on the morphophysiological characteristics of the digestive system, the relative weight of the main digestive organs (pancreas, liver, stomach, and small and large intestine) and the morphology of the villi and crypts in the duodenum, jejunum, ileum, and colon were measured. We chose the jejunum and colon as models for studying intestinal barrier function through occludin concentrations because they are the most representative parts of the small and large intestines, respectively. The jejunum plays a significant role in digestive function as it is the largest portion of the small intestine [27]; in addition, most protein digestion and amino acid absorption occur in the jejunum [28]. The colon represents the largest area for microbial fermentation processes and the absorption of short chain volatile fatty acids (SCFAs) [29]. Therefore, we considered them the sections of the GI tract most susceptible to permeability disruption in the event of intestinal health issues.

#### 2.2.1. Euthanization and Sample Collection

On day 14 post-weaning, six piglets from each experimental group were randomly selected for euthanasia, and samples were collected. The animals were desensitized via CO_2_ inhalation for three minutes, and the jugular vein was sectioned for exsanguination. Subsequently, the abdominal cavity was opened to extract the liver, pancreas, stomach, and small and large intestines, which were then emptied of their contents, washed, and weighed. The relative weights (g/kg live weight) were calculated. The contents of the stomach, duodenum, jejunum, ileum, cecum, and colon were measured using a potentiometer (HANNA^®^ instruments pH 211, Cluj-Napoca, Romania). Samples of approximately 10 cm were collected from the duodenum, jejunum, ileum, and colon and preserved in buffered formalin (10%) until processing. A 2 cm wide area of the mucosa was fixed in paraffin, and 5 µm sections were cut and stained with hematoxylin–eosin to determine the depth of the crypt (from the villi base to the bottom of the crypt) (µm) and height (from apex to base) of the villi (µm). Approximately 10 villi were examined under an optical microscope with a 10× objective using a Primo Star optical microscope (Carl Zeiss, Oberkochen, Germany) [30].

#### 2.2.2. Presence of Occludins

To determine the presence of occludins, 10 cm of jejunum and colon were obtained from three piglets in each experimental group and selected at random. Histological sections were prepared to produce antigen–antibody reactions and quantify occludins by adapting the immunohistochemistry technique, which is briefly described below. The tissues were incubated with a polyclonal anti-occludin antibody (Thermo Fisher Scientific Bioss Brand, Waltham, MA, USA), Catalog No.:BS-1495R, diluted to 1:100 and horse serum (1:1000). The tissues were subsequently incubated with a Goat anti-Rabbit IDG (H+L) (1:60) secondary antibody (Brand Thermo Fisher Scientific Invitrogen, Catalog No.: 32460). Finally, staining was achieved using 3,3′-Diaminobenzidine tetrahydrochloride hydrate (Thermo Scientific Chemicals) 1:8000 *w*/*v* and H_2_O_2_ 1:2500 in phosphate buffer saline (PBS). This reaction produced a sepia precipitate in the immunoreactive cells. The samples were fixed with entellan (a rapid mounting medium for microscopy). Photographic analyses of three different sections of the jejunum and colon were performed on each slide. The samples were observed under a Primo Star microscope (Carl Zeiss, Oberkochen, Germany) and photographed at 40× and 10× magnifications using the ZEN program, version 1.54k (Carl Zeiss, Jena, Germany). The photographs were analyzed using ImageJ software IJ version 1.46r developed by Wayne Rasband at the U.S. National Institutes of Health (NIH), which can be downloaded from https://imagej.nih.gov/ij (accessed on 5 September 2022).

### 2.3. Coliform and Lactobacillus Populations

Immediately before slaughter, approximately 5 g of feces was collected directly from the anus of each animal for microbiological analysis. The samples were serially diluted and plated onto 100 µL surface plates. To quantify the total coliforms, the norm NOM-113-SSA1-1994 was used [31]. Bile and a violet-red agar (Bioxon™) culture medium was used for coliforms, and MRS agar (Difco Laboratories, Detroit, MI, USA) was used for Lactobacillus [32]. The results were expressed as the log of colony-forming units (CFUs) counted per gram of feces.

### 2.4. Incidence and Severity of Diarrhea

The incidence and severity of diarrhea in each pen were evaluated daily. Diarrhea incidence (DI) was measured as the number of days diarrhea was observed within a pen. Diarrhea severity (DS) was defined as a daily visual score of 0–3 [8] based on fecal consistency: 3 = severe, highly fluid diarrhea; 2 = moderate diarrhea; 1 = light, pasty diarrhea; 0 = no diarrhea. Daily scores were added for each week and divided into seven days to obtain the diarrhea severity index for each pen.

### 2.5. Statistical Analysis

In the performance trial, the variables were analyzed according to a random block design, with the initial weight used as a blocking factor and the pen used as the experimental unit. Other variables were analyzed according to a completely randomized design in which piglets were used as the experimental unit. The General Linear Model (GLM) procedure of the statistical software SAS (version 9.2, SAS Inst. Inc., Carry, NC, USA) was used. to perform statistical analysis. The significance of differences between means was determined using Tukey’s test. *p* < 0.05 was considered as significant.

## 3. Results

### 3.1. Piglet Performance

The average daily feed intake (ADFI), average daily gain (ADG), and feed efficiency (FE) of piglets were not affected by the different diets (*p* > 0.05). In the first post-weaning week, all the piglets, regardless of the diet, exhibited reduced feed intake. Consequently, they lost weight and had a negative FE. In the second week, all the animals recovered as their performance improved with an increase in ADFI and ADG, reaching an FE very close to the ideal value of 1 (Table 2).

### 3.2. Intestinal Morphophysiological Characteristics

The relative weights of the digestive organs and the pH of the digestive contents (Table 3) did not differ (*p* > 0.05) between the animals on the different experimental diets.

In the duodenum, the villi were wider (*p* < 0.01) in the piglets fed the SB and PPC-SB diets, and intermediate in the C+ diets (Table 4). Regarding the jejunum, these values were significantly higher (*p* < 0.01) in the C+ and PPC-SB groups. No *differences* between piglets were observed (*p* > 0.05) in the other intestinal segments.

The group fed a diet with antibiotics (C+), as well as those fed a mixture of the two functional feeds (PP-SB), exhibited higher concentrations of occludins (*p* < 0.01) in the jejunum (Figure 1).

In the piglets that were given feed containing only Sb or PP, there was a lower abundance of occludins, like animals fed the C-diet, but no differences were observed (*p* = 0.5587) in the concentration of occludins in the colon (Figure 2).

### 3.3. Microbiological Analyses and Post-Weaning Diarrhea

Animals fed the antibiotic diet had a lower (*p* < 0.05) number of coliforms in their feces when compared to the other animals. However, the intake of the PPC diet reduced the proportion of these bacteria, leading to effects like those observed in piglets that fed the antibiotic diet. The feces samples from piglets that consumed SB and PPC-SB contained more *Lactobacillus* (*p* < 0.001) than those that consumed the diets C−, C+, and PPC (Table 5).

Regarding post-weaning diarrhea, all piglets, regardless of the diet they consumed, presented diarrhea (*p* > 0.05) during the experimental period (Table 5), albeit with mild to moderate severity. On average, the incidence was lower in week 1 than in week 2 (4.5 vs. 6.5), whereas there was a slight increase in severity in week 2 (1.7 vs. 1.8).

## 4. Discussion

The combined effects of different stress factors induce functional and structural changes in the gastrointestinal tract during the weaning period, resulting in low feed intake, poor initial growth, weight loss, and diarrhea [4]. Furthermore, weaning is associated with the increased expression of inflammatory cytokines in the intestine, which can contribute to anatomical and functional intestinal disorders [33].

The piglets’ weight loss observed in this study could therefore be considered “normal” or “expected”, suggesting that the presence or absence of antibiotics or functional feeds in the diets was insufficient to overcome the problems generated by weaning, resulting in the low performance observed in piglets during the first week post-weaning. However, piglets must adapt quickly to these changes during the post-weaning period, which is why in the following week, a significant increase in ADFI was observed, increasing from an average of 101.4 g/day in the first week to 261 g/day in the second week; that is, the piglets increased their feed intake 2.29-fold. Consequently, the ADG also significantly increased, with the animals progressing from an average negative weight gain of −22 g/day across all treatment groups to 202 g/day.

In another study [19], animals treated with *Saccharomyces boulardii* RC009 in the first two weeks post-weaning showed no differences in their ADG compared to the untreated group; however, the probiotic influence on piglets occurred during the fourth post-weaning week. It is possible we concluded our experiment before the functional foods could exert their benefits on growth performance.

The relative weights of the digestive organs were slightly higher, except for the stomach, which had a higher relative weight (9.4 g/kg) compared to those observed in previous studies with piglets of the same origin [34]. At the macroscopic level, the relative weights of the digestive organs indicated that the harmful effects of weaning were not detrimental to visceral development. Under limited protein conditions, the amino acid requirements of the gut remain relatively high, and there is a preference to maintain the size of the digestive organs at the expense of other tissues, as protein deposition in the gastrointestinal tract is prioritized [35].

However, the results observed at the microscopic level, such as the greater width of the villi in the duodenum and greater size in the jejunum, revealed the positive effect of the functional foods PP and Sb, specifically when they were provided in combination and without antibiotics. This contributed to an increase in the digestion and absorption surface in these intestinal segments [27,28], counteracting the atrophy of the intestinal structure observed during the post-weaning phase [4,36].

The greater abundance of occludins in the jejunum observed in animals in the PPC-SB experimental treatment confirms the positive effect of the combination of these two functional foods on intestinal structure. Although the effects of probiotics on the intestinal barrier are commonly studied in rodents, reports on pigs, especially in piglets, are scarce [37]; however, some authors have shown that the use of yeast in piglet diets increased the abundance of occludins in the jejunum [38]. These proteins play central roles in sealing intercellular spaces [39] because they are essential for the selective permeability of enterocytes [40], as integral membrane proteins with four transmembrane domains located at the tight junctions [41]. They therefore serve as a barrier for the passage of undesirable substances through the paracellular spaces of enterocytes, maintaining intestinal integrity and preventing harmful agents like toxins and allergens [21], while microorganisms can pass through the mucosa, inducing inflammatory and immunological responses [23]. The integrity of the intestinal barrier is complex, and the collectivity of the elements that compose it (structure of tight junctions, mucus, microbiota, and immune system) maintains the health of animals at the local and systemic levels [38]. Our microscopic results are relevant since maintaining the integrity of the intestinal villi and the junctions between the cells that cover the villi should be an essential function of the feed ingested by newly weaned piglets, as changes in histological parameters are one of the primary causes of decreased digestion and absorption of nutrients, which contribute to post-weaning diarrhea [42].

Our results indicated that diarrhea was not associated with a specific diet, as all piglets presented with this digestive disorder during the experimental period. This is probably due to the low digestive capacity of piglets and the lack of adaptation of the intestine to the new diets, which leaves many dietary substrates, especially those of protein origin, to be metabolized by the microbiota of the large intestine. This can increase the levels of potentially toxic substances, such as branched-chain fatty acids, ammonia, biogenic amines, hydrogen sulfide, and phenolic and indolic compounds reported by different studies [43], which have been implicated in the pathogenesis of post-weaning diarrhea. In this study, diarrhea was of moderate severity and did not compromise the health of the animals, who managed to recover their body weight in the second week post-weaning, and the development of the digestive organs was not affected.

Previous studies support the idea that functional foods can help reduce the incidence of diarrhea in piglets during the post-weaning phase [8]. However, the results vary between authors, and it is not conclusive. Some of them do not report as significant a reduction in diarrhea as we reported in this work [19,44,45].

A sudden change in feed causes bacterial dysbiosis in the GIT with increased coli-form populations, which may explain the more significant quantities of CFUs of coliform/g feces observed in the animals [5]. The abrupt reduction in *Lactobacillus* spp. at weaning, as mentioned by other authors [5,46], was also observed in animals fed the C− and PPC diets. In the cases of those fed Sb alone or a combination of PP and Sb, a more significant number of CFUs of Lactobacillus/g of feces and a greater *Lactobacillus*/coliform ratio (0.79:1, 0.70:1, and 0.47:1 for the animals) were observed for the SB, PPC-SB, and C− diets, respectively. The inclusion of antibiotics and, to a lesser extent, PP controlled this increase in coliforms and had a slight effect on *Lactobacillus* spp. (*Lactobacillus*/coliform ratio of 0.72:1). The combined effects of PP and Sb may significantly contribute to intestinal health. This is likely due to the antimicrobial peptides found in PPC [9,14] that help control pathogenic bacteria, alongside the benefits of Sb, which is known to enhance antioxidant activity, promote anti-inflammatory responses, and improve the balance of intestinal microbiota [47]. Together, these factors can lead to a healthier gastrointestinal environment, supporting overall digestive health and potentially reducing the risk of gastrointestinal diseases. However, this mechanism requires further verification, and the presence of these antimicrobial peptides in PP needs to be confirmed.

Due to the functional properties of potato proteins, as described by different authors, better results were expected in animals that consumed PPC added individually to the diet. A probable explanation for this is that during the procedure for the extraction, the separation and purification of PP were structural changes in these proteins, and they consequently lost part of their functionality. High-performance precipitation methods are used to extract proteins from potato juice; however, these can alter their functional properties, leading to partial or total loss of the protein functionality, which restricts the application of PPC in animal feed [48]. Recently, there has been increased interest in PP, and many authors have investigated alternative treatments and the modification of technologies used to eliminate this threat [11,49]. In future studies, a prior evaluation of the functionality of the PPC proteins is recommended.

Fortunately, the combination of PP and Sb had positive results for intestinal health due to the better height of the jejunum villi and the higher concentration of occludins in this tissue and *Lactobacillus* in the feces of the piglets that consumed the mixture of these two functional foods. Other researchers also observed the ability of *Saccharomyces* to associate with other functional foods to generate benefits to the host. Piglets fed a dietary supplement with blood plasma and *Saccharomyces cerevisiae* over 0–21 days exhibited a significant impact on the growth performance, nutrient digestibility, fecal microbial, and gas emission at the end of the experiment [44]. On the other side, the combination of *Pediococcus pentosaceus* RC007 (a new probiotic) and *Saccharomyces boulardii* RC009 promoted healthier gut microbiota, with the reduced abundance of Proteobacteria and Cyanobacteria, and could be an effective substitute for antibiotics in improving pig production performance [45]. Moreover, the use of a composite probiotic consisting of *Lactobacillus plantarum, Lactobacillus reuteri, and Bifidobacterium longum* showed that the mean level of ZO-1, another tight junction protein, was significantly higher in the probiotic-treated group compared to the control group [50].

## 5. Study Limitations and Future Research

Most studies in this field have investigated alternatives to antibiotics for addressing gastrointestinal health problems in the post-weaning phase, either offering isolated alternatives or combining different elements within the same category, such as a combination of different probiotic strains. However, the results of this study highlight the value of combining beneficial effects from different categories of functional foods, rather than using them in isolation. Nevertheless, further research should aim to corroborate this hypothesis and elucidate the mechanism(s) behind the combined effects of PP and Sb observed in this study.

In future research, we recommend investigating the effect of diets supplemented with PP and Sb on the concentration of SCFAs from fermentative processes in the GIT, especially butyric acid in the colon. This SCFA plays a particular role in maintaining and restoring tight junctions and the intestinal barrier [51].

Several authors agree that the most critical phase following weaning is the first two weeks; however, this duration may not be sufficient to fully assess the long-term effects of supplementation. Extending the study could provide a more comprehensive understanding of its benefits. It would be interesting to determine whether the positive results observed at the end of the second week post-weaning persist or improve in the following two or three weeks.

## 6. Conclusions

It was concluded that including a combination of PP and Sb in the diet without antibiotics did not negatively affect performance or the development of digestive organs during the post-weaning period. The PP-Sb combination prevented the atrophy of jejunal villi, increasing their length, and had an effect similar to that of antibiotics. *Saccharomyces boulardii*, alone or in combination with PP, stimulated an increase in Lactobacillus. Simultaneously, the addition of PPC and Sb to the piglet diet promoted enhanced adhesion between enterocytes in the jejunum, similar to antibiotics; reduced intestinal permeability; and protected the function of the intestinal barrier. Moreover, it reduced the likelihood of microorganisms penetrating enterocytes and entering the bloodstream, potentially causing systemic infections. Including these functional feeds in piglet diets contributed to maintaining intestinal health during the critical post-weaning phase.

## Figures and Tables

**Figure 1 animals-15-00985-f001:**
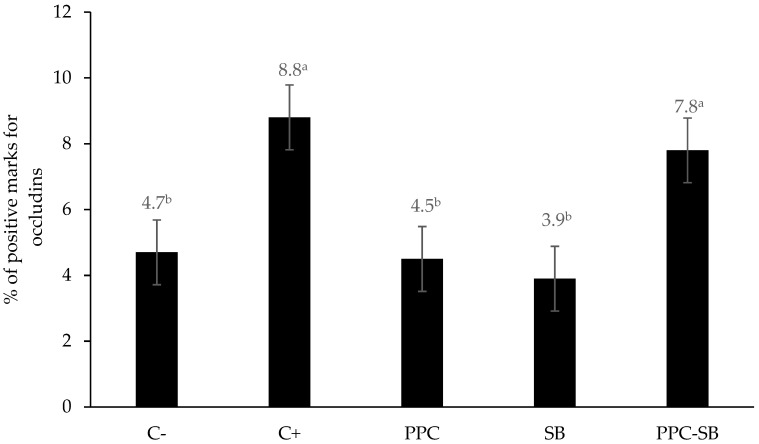
Concentration of occludins in the piglet’s jejunum. ^a,b^ Values of bars with different letters differ significantly (*p* < 0.05). C−: diet without antibiotics. C+: diet supplemented with antibiotics. PPC: diet with PP. SB: diet with Sb. PPC-SB: diet with PP and Sb.

**Figure 2 animals-15-00985-f002:**
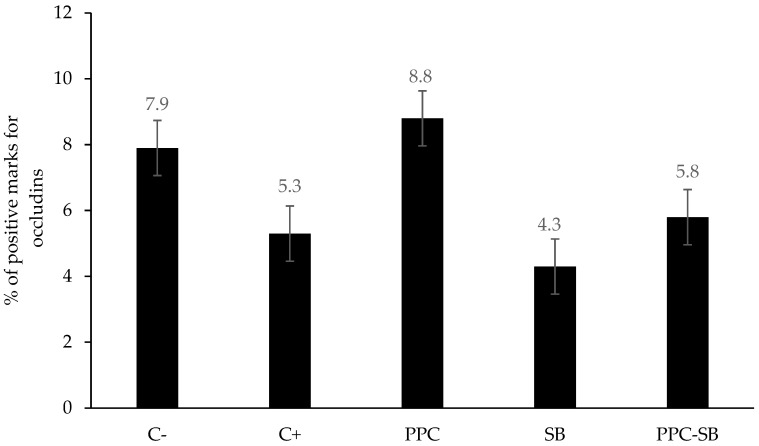
Concentration of occludins in the piglet’s colon. C−: without antibiotics, PP, or Sb. C+: with antibiotics. PPC: with PP. SB: with Sb. PPC-SB: with PP and Sb.

**Table 1 animals-15-00985-t001:** Centesimal composition of the experimental diets.

Ingredients (%)	Experimental Diets				
	C−	C+	SB	PPC	PPC-SB
Maize	44.7	44.56	44.68	43.78	43.77
Soybean Meal	12	12	12	12	12
Soybean Isolate	8.32	8.34	8.33	3.74	3.74
Potato Protein Concentrate				6	6
Antibiotic ^1^		0.05			
Yeast ^2^			0.01		0.01
Menhaden Fish Meal	5	5	5	5	5
Sweet Whey Milk	24.69	24.69	24.69	24.69	24.69
Maize oil	2.45	2.52	2.45	2.23	2.23
L-Lysine HCl	0.4	0.4	0.4	0.27	0.27
L-Threonine	0.12	0.12	0.12	0.02	0.02
DL-Methionine	0.19	0.19	0.19	0.14	0.14
L-Tryptophan	0.03	0.03	0.03	0.03	0.03
L-Valine	0.02	0.02	0.02		
Salt	0.4	0.4	0.4	0.4	0.4
Calcium Carbonate	0.6	0.6	0.6	0.54	0.54
Dicalcium Phosphate	0.61	0.61	0.61	0.69	0.69
Titanium Dioxide	0.3	0.3	0.3	0.3	0.3
Vitamins Premix ^3^	0.07	0.07	0.07	0.07	0.07
Minerals Premix ^4^	0.1	0.1	0.1	0.1	0.1
Chemical Composition					
DM (%) ^5^	90.2	90.4	90.2	90.9	90.2
CP (%) ^5^	19.9	20.3	20.5	20.0	19.9
EE (%) ^5^	5.35	5.41	5.18	5.35	5.18
ME (Mcal/kg) ^6^	3.400	3.400	3.400	3.400	3.400
NDF ^5^	6.14	6.12	6.28	6.14	6.28

C−: without antibiotics, PP, or Sb. C+: with antibiotics. PPC: with PP. SB: with Sb. PPC-SB: with PP and Sb. ^1^ Linco Spectin premix: 2.2 g lincomycin, 2.2 g spectinomycin (Zoetis, USA). ^2^
*Saccharomyces boulardii* (Levucell SB, CNCM I-1079, Lallemand Animal Nutrition, Canada). ^3^ Per kg feed: CaCO_3_, 1.43 mg; CuSO_4_·H_2_O, 55.5 mg; FeSO_4_·H_2_O, 333.3 mg; C_2_H_8_N_22_HI, 1.01 mg; MnSO_4_·H_2_O, 135 mg; Na_2_SeO_3_, 0.5 mg; ZnSO_4_·H_2_O, 338 mg. ^4^ Per kg feed: vitamin A, 6563 IU; vitamin D_3_, 893 IU; vitamin E, 33.5 IU; vitamin K, 1.2 mg; riboflavin, 3.6 mg; vitamin B12, 18 μg; choline, 356 mg; niacin, 17 mg; pantothenic acid, 14 mg; thiamine, 1.4 mg; pyridoxine, 2.8 mg; biotin, 127 μg; folic acid, 0.9 mg. ^5^ Analyzed value. ^6^ Calculated value.

**Table 2 animals-15-00985-t002:** Piglets’ performance parameters.

Items	Experimental Diets					*p*	SEM
	C−	C+	SB	PPC	PPC-SB		
ADFI (g/day)							
Week 1	105	96	98	103	105	0.179	3.1
Week 2	255	258	256	265	271	0.277	5.3
ADG (g/day)							
Week 1	−28	−21	−30	−13	−15	0.428	4.4
Week 2	210	230	215	219	224	0.829	5.4
FE							
Week 1	−0.270	0.249	−0.345	−0.194	−0.191	0.684	0.052
Week 2	0.821	0.914	0.835	0.830	0.833	0.702	0.015

C−: without antibiotics, PP, or Sb. C+: with antibiotics. PPC: with PP. SB: with Sb. PPC-SB: with PP and Sb. ADFI: average daily feed intake. ADG: average daily gain. FE: feed efficiency. *p*: probability. SEM: standard error of the mean.

**Table 3 animals-15-00985-t003:** Morphophysiological characteristics of organs of the digestive system.

Items	Experimental Diets					*p*	SEM
	C−	C+	SB	PPC	PPC-SB		
Body weight (kg)	8.100	8.527	7.715	8.238	7.933	NS	0.177
Relative body weight (g × BW^−1^)							
Pancreas	2.2	2.1	1.9	2.3	2.1	0.329	0.065
Liver	29	32	30	33	28	0.326	0.818
Stomach	8.8	8.1	7.0	7.9	7.9	0.581	0.312
Small Intestine	61	57	58	65	60	0.279	1.285
Large Intestine	17.7	19.8	18.1	18.2	19.8	0.863	0.794
pH of Contents							
Stomach	2.8	3.7	4.1	3.5	3.7	0.216	0.163
Jejunum	5.4	5.5	5.6	5.5	5.7	0.510	0.064
Ileum	5.9	6.0	6.1	5.8	5.9	0.823	0.081
Ceacum	5.4	5.4	5.4	5.4	5.5	0.973	0.061
Colon	5.5	5.5	5.5	5.4	5.6	0.802	0.047

C−: without antibiotics, PP, or Sb. C+: with antibiotics. PPC: with PP. SB: with Sb. PPC-SB: with PP and Sb. *p*: probability. SEM: standard error of the mean. NS: non-significant.

**Table 4 animals-15-00985-t004:** Morphology of intestinal villi and crypts.

Items	Experimental Diets					*p*	SEM
	C−	C+	SB	PPC	PPC-SB		
Duodenum							
VH (μm)	344	369	363	331	375	0.218	6.8
VW (μm)	115 ^bc^	130 ^ab^	138 ^a^	108 ^c^	138 ^a^	0.009	2.7
CD (μm)	197	223	213	227	240	0.717	8.3
Jejunum							
VH (μm)	378 ^b^	439 ^a^	337 ^b^	343 ^b^	435 ^a^	0.003	8.0
VW (μm)	112	123	111	94	109	0.642	4.9
CD (μm)	236	247	202	244	234	0.212	6.7
Ileum							
VH (μm)	298	331	332	366	361	0.303	9.9
VW (μm)	108	111	107	117	116	0.782	3.1
CD (μm)	208	187	194	203	214	0.437	5.3
Colon CD (μm)	320	312	305	299	337	0.374	6.3

^a,b,c^: Values of means with different letters differ significantly (*p* < 0.05). C−: without antibiotics, PP, or Sb. C+: with antibiotics. PPC: with PP. SB: with Sb. PPC-SB: with PP and Sb. VH: villus height. VW: villus width. CD: crypt depth. SEM: standard error of the mean. *p*: probability.

**Table 5 animals-15-00985-t005:** Microbiological analyses of feces; diarrhea incidence and severity.

Items	Experimental Diets					*p*	SEM
	C−	C+	SB	PPC	PPC-SB		
Coliforms (CFUs/g)	7.4 ^a^	5.7 ^b^	7.5 ^a^	7.2 ^ab^	8.1 ^a^	0.040	0.23
*Lactobacillus* (CFUs/g)	3.5 ^b^	4.1 ^b^	5.9 ^a^	3.8 ^b^	5.7 ^a^	0.001	0.19
DI							
Week 1 (days)	4.6	4.1	4.3	4.9	4.8	0.386	0.14
Week 2 (days)	1.8	1.7	1.6	1.7	1.8	0.894	0.05
DS							
Week 1	6.3	6.5	6.6	6.5	6.4	0.927	0.12
Week 2	1.7	1.9	1.9	1.9	1.7	0.539	0.06

^a,b^: Values of means with different letters differ significantly (*p* < 0.05). C−: without antibiotics, PP, or Sb. C+: with antibiotics. PPC: with PP. SB: with Sb. PPC-SB: with PP and Sb. CFUs: colony-forming units. DI: diarrhea incidence. DS: diarrhea severity. SEM: standard error of the mean. *p*: probability.

## Data Availability

The data supporting this study will be shared upon reasonable request to the corresponding authors.

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
