# Peer review of "Supplementation with Potato Protein Concentrate and Saccharomyces boulardii to an Antibiotic-Free Diet Improves Intestinal Health in Weaned Piglets"

_animals, 2025, doi:10.3390/ani15070985_

Round 1
Reviewer 1 Report
Comments and Suggestions for Authors
Dear Authors,
the manuscript " Supplementation with Potato Protein Concentrate and Saccharomyces boulardii to an antibiotic-free diet improves intestinal health in weaned piglets" (animals-3529852) by the authors: Tércia Cesária Reis de Souza, Gerardo Mariscal Landín, Ulisses Moreno Celis, Teresita Hijuitl Valeriano, José Guadalupe Gómez-Soto, Christian Narváez Briones, is related to the important field of animal biology, health and nutrition. It is aimed as the following: “to evaluate the effect of isolated or combined intake of potato protein concentrate (PPC) and Saccharomyces boulardii (SB) on productive performance and some morphophysiological responses of the digestive tract in the two post-weaning weeks”. About 132 pigs (Fertilis × Genetiporc) were weaned at 19.8±1.6 days and weighed in at 6.2±0.85 kg. It is positive that the authors include the “centesimal composition of the experimental diets” in the separate table (Table 1 at the line 124) using the principle of Ideal Protein, using the Standardized Ileal Digestibility of amino acids. The piglets were assigned to one of the five experimental diets according to a randomized complete block design, with initial weight used as a blocking factor: C- (negative control, without PPC or SB), C+ (positive control, with antibiotic), SB (without antibiotic, with SB), PPC (without antibiotic, with PPC), or PPC-SB (without antibiotic, with PPC + SB). The results were focused at the following parameters: the average daily feed intake (ADFI), average daily gain (ADG), feed efficiency (FE) of piglets (Table 2 “Piglets’ performance parameters”); morpho-physiological characteristics of the digestive system (Table 3); morphology of intestinal villi and crypts (Table 4); bacteria population, as well as, diarrhea incidence and severity (Table 5). Thus, including a combination of PPC and SB in the diet without antibiotics did not negatively affect the performance or development of digestive organs in the post-weaning period. The PPC-SB combination prevented atrophy of the jejunal villi, increasing their length, and had a similar effect as the antibiotics. Saccharomyces boulardii, alone or in combination with PPC, stimulated an increase in Lactobacillus. Simultaneously, the addition of PPC and SB to the piglet diet promoted outstanding adhesion between enterocytes in the jejunum, similarly to antibiotics, reduced intestinal permeability, and protected the function of the intestinal barrier. Likewise, it reduced the probability that microorganisms would penetrate enterocytes and enter the blood, causing systemic infections. Therefore, including these functional feeds in piglet diets contributed to maintaining intestinal health in the critical post-weaning phase.
I do not doubt the technical quality of the work and feel that there is a sufficient impact on a broader readership to justify publication in the "Animals". This topic is in the frame of the journal scope, the subject matter is treated in depth. Thus, the present manuscript is actual and important, especially in the field of the animal health and nutrition.
There are some comments:
- There are some similar segments in the Simple Summary and the Abstract. For example, Lines 13-15 and 27-29: “we included in the diet two functional foods (potato protein concentrate (PPC) and Saccharomyces boulardii (SB)) isolated or combined and evaluate their effects on productive performance, integrity of the small intestine, and weight of the digestive organs in the post-weaning period”; Lines 20-22 and 32-34 “The piglets fed the combination of PPC and SB had a higher concentration of the occludin proteins responsible for the union between the cells of the small in testine”. Please, modify these similar segments.
- The part 1. “Introduction”. It is important to relate each reference (from the following row [10-13]) to the particular word in the line 66: “… like antiallergic [!?], antioxidant [!?], and anticancer activity [!?]…” instead of the general citation [10-13].
- The part 2. “Materials and Methods”. The amount of the animals in each of the five experimental group is not defined. It is important to define exactly.
There are no comments for the parts: 3. “Results”, 4. “Discussion”, 5. “Conclusions”. The presence of the “Abbreviation” section in the first or the last parts of this manuscript can be valuable.
Author Response
Comments 1: There are some similar segments in the Simple Summary and the Abstract. For example, Lines 13-15 and 27-29: “we included in the diet two functional foods (potato protein concentrate (PPC) and Saccharomyces boulardii (SB)) isolated or combined and evaluate their effects on productive performance, integrity of the small intestine, and weight of the digestive organs in the post-weaning period”; Lines 20-22 and 32-34 “The piglets fed the combination of PPC and SB had a higher concentration of the occludin proteins responsible for the union between the cells of the small intestine”. Please, modify these similar segments.
|
Response 1: I agree with this comment. I have made the recommended modifications in the simple summary and in the summary page 1, lines: 12-18 and 20-35, respectively.
|
Comments 2: The part 1. “Introduction”. It is important to relate each reference (from the following row [10-13]) to the particular word in the line 66: “… like antiallergic [!?], antioxidant [!?], and anticancer activity [!?]…” instead of the general citation [10-13]. |
Response 2: I agree with this comment. I have made the separation of references 10 - 13 and put them with their respective action. Please, see page 2, lines 67-68 The text was as follows: “functional benefits for human health like antiallergic [10], antioxidant [11,12], and anti-cancer activity [11,13] which can also cover animal health.” ” |
Comments 3: The part 2. “Materials and Methods”. The amount of the animals in each of the five experimental group is not defined. It is important to define exactly. Response 3: I agree with this comment. I define it. You can see it on page 3, lines 115-116, marked in yellow in the manuscript. The text was as follows: “…6.2 ± 0.85 kg, and divided into five groups, two with 27 piglets and three with 26 piglets consuming, respectively, the experimental diets….” 4. Response to Comments on the Quality of English Language |
Point 1: No comments on the quality of English Language by Reviewer 1 |
Response 1: Even without any comments, I submitted it to the rapid English review in MDPI Author Services to improve the quality of English Language. |
5. Additional clarifications |
Additionals comments by the Reviewer 1. [There are no comments for the parts: 3. “Results”, 4. “Discussion”, 5. “Conclusions”. Thank you very much for taking the time to review this manuscript. [The presence of the “Abbreviation” section in the first or the last parts of this manuscript can be valuable.] Thank you very much for your comment. I prepared this list and I may add it to the manuscript. The abbreviations marked yellow were changed in the new version of the manuscript to differentiate supplemented functional foods from the diets used, and to make the text more understandable. |

Reviewer 2 Report
Comments and Suggestions for Authors
The search for new dietary alternatives becomes essential to obtain advances in animal production througt better nutrition. Tis research sets an important precedent regarding the intestinal health of pigs that goes hand in hand with optimal digestion and utilizacion of nutrints. Below is a corrections and suggestions to improve the quality of the manuscript.

It is suggested to seek assistance from an expert to improve the writing and grammar of the manuscript to improve its quality
Author Response
Comment 1: Lines 4-11: Correct author affiliations (and numbers).
Response 1: Thank you for your comment. We appreciate your observation. We agree. Lines
Comment 2: Line 17: it is suggested to clarify the explanation of the types of diets used in the study, specifically the total number of diets and the control diet.
Response 2: Thank you for your observation. We agree. We clarify how the diets are on lines 26-27.
Comment 3: Line 23: the term "post-weaning" is more utilized that "after the weaning"
Response 3. Thank you for your observation. We agree. We change into different parts after weaning to post-weaning: lines: 14, 18, 34, 130, 225, 263, 296, and 297.
Comment 4: Line 29: is not clear if the analysis (or sampling) carried out during the two weeks post-weaning or after two weeks post-weaning (improve writing)
Response 4: Thank you for Comment. We agree. We changed the wording to “after two weeks post-weaning”. Line 25.
Comment 5: Lines 29-30: add (C-) and (C+) to the explanation of diets. These acronyms are later used
Response 5: Thank you for your comment. We agree. We added (C-) and (C+) to the explanation of diets. Lines 29-30.
Comment 6: Line 33: in the intestinal villi of jejunum? (in the jejunum and intestinal villi)
Response 6: Thank you. We appreciate your observation. We corrected this error. Lines 29-30.
Comment 7: Line 45: the concept of nutritional means is not understood
Response 7: Thank you. We appreciate your observation. We changed the word “means” to alternatives to be more understandable. Line 44.
Comment 8: Line 64: use the acronym for gastrointestinal tract (GIT)
esponse 8: Thank you. We appreciate your observation. We used the acronym GIT. Lines 45, 67, 349, and 400.
Comment 9: Line 71: can you explain the "productive criteria" concept?
Response 9: Thank you for your comment. We appreciate your observation. We are referring to zootechnical parameters, so we changed the text. Line 84.
Comment 10: Line 111: The explanation of cage metrics should be improved.
Response 10: Thank you for your comment. We appreciate your observation. We changed the text:
“The piglets were housed in pens that were raised 38 cm above the ground and measured 115 cm wide and 150 cm long, making a total surface area of 1.7 m2”. Lines 130-132.
Comment 11: Line 108: The number of animals used in the study cannot be approximated. ("a total of 132 piglets...")
Response 11: Thank you for your comment. We appreciate your observation. We corrected this error. Line 114.
Comment 12: Line 108: "Fertilis × Genetiporc" corresponds to the cross between two breeds? Or do they correspond to the genetic companies from which the animals precede? Clarify.
Response 12: Thank you for your comment. We appreciate your observation. It is the genetic line of pigs. Line 114.
Comment 13: Line 124: Diets do not have AQP.
Response 13: Thank you for your comment. We agree. The data on the chemical composition (DM, CP, EE, ME and NDF) of the diet has already been attached to table 1.
Comment 14: Lines 125-192-201-220-229-242: the acronym of negative control diet are incomplete (add "-" to "C")
Response 14: Thank you for your comment. We agree. It has already been completed in all indicated places. Lines 139, 231, 238, 245, and 276.
Comment 15: Line 128: It is not necessary to put the word "yeast" if you put the numerical
superscript.
Response 15: Thank you for your comment. We agree. The word yeast was removed. Line 139.
Comment 16: Line 141: potentiometer (manufacturing, country)
Response 16: Thank you for your observation. The potentiometer manufacturer and country of origin (HANNA® instruments pH 211, Romania) have already been entered. Line 168.
Comment 17: Line 148: Why is the jejunum and colon used to determine the presence of Occludins and no other tissue? Are there any associated scientific criteria?
Response 17: Thank you for your comment. We appreciate your observation. We chose jejunum and colon because they are the most representative parts of the small and large intestines, respectively. As the largest part of the small intestine, the jejunum is the most important part for enzymatic and mechanical digestion processes, and the largest surface area for nutrient absorption. The colon represents the largest area for microbial fermentation processes and the absorption of short-chain volatile fatty acids. So, we think they are also the parts most susceptible to problems with permeability disruption in case of intestinal health risks. We mentioned the importance of the jejunum in the lines 153-157.
Comment 18: Line 154: "Goat and Rabit" IDG (H+L) that's mean Goat anti Rabbit?
Response 18: We appreciate your observation. Indeed, it was a Goat anti-Rabbit antibody. We have corrected the grammatical error. Lines 184-189.
Comment 19: Additionally, the diluted concentration of a single antibody is indicated.
Response 19: Thank you for your observation. We have added the concentration used on line 184.
Comment 20: Line 155: the paragraph "Finally, it was performed using 3,3'-Diaminobenzidine tetrahydrochloride hydrate" it is not explanatory.
Response 20: Thank you for your observation. We have corrected what you indicated. Lines 186-187. We did not include too many details about the technique since it is widely used.
Comment 21: Line 158: what mean "entelan"?
Response 21: Thank you for your observation. We were referring to Entellan (rapid mounting medium for microscopy). Line 189.
Comment 22: Line 194-195: the paragraph "1Linco Spectin premix: 2.2 g lincomycin, 2.2 g spectinomycin (Zoetis, USA). 194 2Yeast: Saccharomyces boulardii (Levucell SB, CNCM I-1079, Lallemand Animal Nutrition, Canadá)" does not apply
Response 22: Thank you for your comment. We agree. We removed it. Lines 231-233.
Comment 23: Line 206: it is suggested change "With regards" for "regarding"
Response 23: Thank you for your comment. We agree. We made the indicated change. Line 241.
Comment 24: Line 262: "average daily gain (ADG)" only use the acronym
Response 24: Thank you for your comment. We agree. We made the indicated change in line 296.
Comment 25: Lines 268-269: "digestive" and "gastro-intestinal". Use the acronym of the latter (GIT)
Response 25: Thank you for your comment. We agree. We made the indicated change in line 396.
Comment 26: Line 278: What do you mean whit "the most considerable portion of the small intestine" the largest or the most important?
Response 26: Thank you for your comment. We appreciated your observation. We believe both. For the digestion process, the jejunum is the most important part, mainly of dietary proteins, as it represents the largest absorption surface of amino acids in non-ruminant animals. We changed the writing 309.
Comment 27: Line 287: add quote to the next paragraph "Therefore, they serve as a barrier for the passage of undesirable substances through the paracellular spaces of enterocytes, maintaining intestinal integrity and preventing harmful agents and microorganisms from passing through them."
Response 27: Thank you for your comment. We appreciated your observation. We added quote [23] in Line 325.
Comment 28: Line 317: the "C" diet (do you refer to C+ or C- diet?)
Response 28: Thank you for your comment. We appreciated your observation. We refer to C-. We changed the text. Line 352.
Comment 29: Lines 458-465: remove instructions for writing references
Response 29: Thank you for your comment. We appreciated your observation. We removed them.
Simple summary: seems to be the "summary of the abstract", it is suggested to rewrite a new simple summary that incorporates the essentials of the study in a different, simple and well explanatory way.
Response: Thank you for your comment. We appreciated your observation. We rewrite a simple summary based on the comments of all reviewers. Lines 12-18.
Abstract: The reported results do not have an associated p value. It is suggested to improve the conclusion of the study at the end of the section since it does not clearly differ from the results
Response 30: Thank you for your comment. We appreciated your observation. We improved the abstract based on the comments from all the reviewers and the results obtained. We will increase and improve our conclusions. Lines 20-35.
Introduction: this section shows good bibliographic support. Some ideas in the text are not clear, which could be due to problems in the writing.
Response 31: Thank you for your comment. We appreciated your observation. We made some changes in different places to improve our writing.
Materials and methods:
Comments 32. This section explains the five experimental diets well, but it is not clear how the piglets were grouped into blocks according to their weight. The subheadings of the material and methods section are lost within the text. They could be put in a separate point and italicized.
Response: Thank you for your comment. We appreciated your observation. We changed the text to clarify how the piglets were grouped into blocks according to their weight. Lines 114-116. Moreover, we put the subheadings of the material and methods section in a separate point and italicized.
Results:
Comment 33. Table 2: Why didn’t they specify the no significative p-values in the table?. In the diets tittles only “C-“ it is in bold. Incorporates p-value in some results and not in others (line 186 and line 199). Unify writing form of show p-value (P> or p>), for example in lines 205-207-208)
Thank you for your comment. We agree with all the comments. Line 230.
Table 2: Why didn’t they specify the no significative p-values in the table?
Response 33.1: We follow your recommendation and put the p-values in table 2. Line 230.
In the diets tittles only “C-“ it is in bold.
Response 33.2: We follow your recommendation, and we put the titles of all others in bold in table 2. Line 230.
Incorporates p-value in some results and not in others (line 186 and line 199).
Response 33.3: We follow your recommendation and put all the results. Lines 225, 236, 242, 243, 249, 258, and 264.
Unify writing form of show p-value (P> or p>), for example in lines 205-207-208)
Response 33.4: We unified the wording of all the values. Lines 225, 236, 242, 243, 249, 258, and 264.
Comment Results 34: Table 3: Why didn’t they specify the no significative p-values in the table?.
Table 3: Why didn't they specify the no significative p-values in the table?. The title could be better specified "morphophysiological characteristics of parts/organs of the digestive system". In the diets tittles only "C-" and the first value of BW (8100) are in bold
Thank you for your comment. We agree with all the comments.
Table 3: Why didn’t they specify the no significative p-values in the table?
Response 34.1: We put the p-values in table 3. Line 237.
The title could be better specified “morphophysiological characteristics of parts/organs of the digestive system”.
Response 34.2: We changed the title of table 3. Line 237.
In the diets tittles only “C-“ and the first value of BW (8100) are in bold.
Response 34.3: We put the titles of all others in bold in table 3, and we removed the bold from the first value of BW (8100). Line 237.
Comment 35 Results: Table 4: Why didn’t they specify the no significative p-values in the table?. In the diets tittles only “C-“ and Duodenum” are in bold. Letters accompanying significant values should be added in superscript format. Why is Colon only the CD and no VH and VW?, Colon CD are in um or m?.
Response 35: Thank you for your comment. We appreciated your observation.
Table 4: Why didn’t they specify the no significative p-values in the table?.
Response 35.1: We put the p-values in table 4. Line 275.
In the diets tittles only “C-“ and Duodenum” are in bold.
Response 35.2: We put the titles of all others in bold and we removed the bold from the word “duodenum” in table 4. Line 275.
Letters accompanying significant values should be added in superscript format.
Response 35.3: We changed the format of letters accompanying significant values. Line 275.
Why is Colon only the CD and no VH and VW?
Response 35.4: Because in the mucosa of colon there are only crypts and there are no villi.
Colon CD are in um or m?.
Response 35.5: The colon crypts are in mm. Line 275.
Comment 36 Results: Figures 1-2: The titles of the figures are positioned below them. The letter of the figures differs between them
Response 36. Thank you for your comment. We appreciated your observations. We remade the figures and corrected these errors. Lines 251 and 258.
Comment 37 Results: Table 5: Why didn’t they specify the no significative p-values in the table?. In the diets tittles only “C-“, “Coliforms (UFC/g)” and the first value are in bold. Letters accompanying significant values should be added in superscript format. In this table “ID” mean “DI”= Diarrhea incidence and “SD” mean “DS”= Diarrhea severity?. The acronyms used along the writing must be equals. The definitions of these concepts are also not described in the footnote of the table
Thank you for your comment. We appreciated your observations.
Table 5: Why didn’t they specify the no significative p-values in the table?
Response 37.1: We put the p-values on table 5. Line 274.
In the diets tittles only “C-“, “Coliforms (UFC/g)” and the first value are in bold.
Response 37.2: We put the titles of all others in bold and we removed the bold from “Coliforms (UFC/g)” and the first value on table 5. Line 274.
Letters accompanying significant values should be added in superscript format.
Response 37.3: We changed the format of letters accompanying significant values. Line 274.
In this table “ID” mean “DI”= Diarrhea incidence and “SD” mean “DS”= Diarrhea severity?. The acronyms used along the writing must be equals. The definitions of these concepts are also not described in the footnote of the table
Response 37.4: We corrected these acronyms and defined the concepts in the footnote of the table 5. Line 274.
Comment 38 References: quote are written in different formats
Response 38: Thank you for your comment. We appreciated your observation. We corrected the formats of the quotes. Line 274.
Reviewer 3 Report
Comments and Suggestions for Authors
I appreciate the effort put into this research. The study presented in the paper, "Supplementation with Potato Protein Concentrate and Saccharomyces boulardii to an Antibiotic-Free Diet Improves Intestinal Health in Weaned Piglets," contributes to the ongoing search for antibiotic alternatives to mitigate the challenges faced by piglets during weaning.
However, I have a few concerns regarding the research design that should be addressed:
- While the approach is commendable, why did the authors limit the observation period to only two weeks post-weaning? This duration may not be sufficient to fully assess the long-term effects of supplementation. Extending the study could provide a more comprehensive understanding of its benefits.
- How were the Potato Protein Concentrate and Saccharomyces boulardii obtained? Were these commercially available products, or were they synthesized in the laboratory? If the latter, further details on their preparation should be included.
- The Materials and Methods section does not specify which antibiotic was added to the C+ diet. Clarifying this would be beneficial for reproducibility and understanding the comparative effects of the supplementation.
- In Table 5, does the reported coliform count account for the dilution factor? Typically, colony-forming units (CFU) or UFC/g are expressed in scientific notation, as bacterial populations usually exist in large numbers. A clarification on this would be helpful.
- The discussion section lacks insights into potential future research directions. Given that this study only captures a short post-weaning period, it would be valuable to discuss how future studies might explore longer-term effects or alternative supplementation strategies.
- Please remove references Line: 458 and 465 from the list.
- More than 50% of the references are older than five years. It is recommended to incorporate more recent literature to strengthen the study’s alignment with current research trends.
- The summary needs language improvement for clarity. The rest of the article is fine.
Addressing these concerns will enhance the clarity and impact of the study.
Thank you.
Author Response
Comment 1. While the approach is commendable, why did the authors limit the observation period to only two weeks post-weaning? This duration may not be sufficient to fully assess the long-term effects of supplementation. Extending the study could provide a more comprehensive understanding of its benefits.
Response 1: Thank you for your comment. We appreciated your observation. The post-weaning period is thus critical for pig production, health, and performance. This has therefore generated substantial worldwide interest in mitigating post-weaning challenges, and identifying nutritional alternatives, management, and disease prevention to positively influence the structure and function of the gastrointestinal tract (GIT), thereby improving production and health outcomes. Stress at weaning, in addition to the presence of new pathogens, leads to neuroendocrine, immunological, and digestive complications. Lines 41-47.
Comment 2. How were the Potato Protein Concentrate and Saccharomyces boulardii obtained? Were these commercially available products, or were they synthesized in the laboratory? If the latter, further details on their preparation should be included.
Response 2: Thank you for your comment. We appreciated your observation. The Potato Protein Concentrate were commercially available products donated by Grupo NUTEC® (Mex-co) and Lallemand Animal Nutrition (Canada), respectively. Lines 121-122.
Comment 3. The Materials and Methods section does not specify which antibiotic was added to the C+ diet. Clarifying this would be beneficial for reproducibility and understanding the comparative effects of the supplementation.
Response 3: Thank you for your comment. We appreciated your observation. The antibiotic used was Lin-co-Spectin premix, containing 2.2 g of lincomycin and 2.2 g of spectinomycin (Zoetis, USA). Line 122-123.
Comment 4. In Table 5, does the reported coliform count account for the dilution factor? Typically, colony-forming units (CFU) or UFC/g are expressed in scientific notation, as bacterial populations usually exist in large numbers. A clarification on this would be helpful.
Response 4: Thank you for your comment. We appreciated your observation. We put in table 5 the correct units, as you recommend. Line 274.
Comment 5. The discussion section lacks insights into potential future research directions. Given that this study only captures a short post-weaning period, it would be valuable studies might explore longer-term effects or alternative supplementation strategies.
Response 5: Thank you for your comment. We appreciated your observation We added a new heading, “5. Study Limitations and Future Research”, to discuss study limitations and future research, as you recommended. Lines 389-407.
Comment 6. Please remove references Line: 458 and 465 from the list.
Response 6: Thank you for your comment. We appreciated your observation. We removed them.
Comment 7. More than 50% of the references are older than five years. It is recommended to incorporate more recent literature to strengthen the study’s alignment with current research trends.
Response 7: Thank you for your comment. We appreciated your observation. We conducted new bibliographic research and found more recent articles related to the topic. Based on this, we now have 60.78% of references from the last five years (2020-2024).
Comment 8. The summary needs language improvement for clarity. The rest of the article is fine.
Response 8: Thank you for your comment. We appreciated your observation. The manuscript was reviewed by English experts. I hope it has improved. We rewrite the simple summary, and the summary based on the comments of all reviewers. Lines 12-18.

Round 2
Reviewer 2 Report
Comments and Suggestions for Authors We appreciate your willingness to improve the manuscript; we hope to have made a positive contribution.